# lncRNAs UC.145 and PRKG1-AS1 Determine the Functional Output of DKK1 in Regulating the Wnt Signaling Pathway in Gastric Cancer

**DOI:** 10.3390/cancers14102369

**Published:** 2022-05-11

**Authors:** Jung-ho Yoon, Hyojoo Byun, Cristina Ivan, George A. Calin, Dahyun Jung, Sangkil Lee

**Affiliations:** 1Division of Gastroenterology, Department of Internal Medicine, Institute of Gastroenterology, Yonsei University College of Medicine, Severance Hospital, Seoul 03722, Korea; gjukminam@yuhs.ac (J.-h.Y.); hyojoo6870@yuhs.ac (H.B.); 2Department of Experimental Therapeutics, The University of Texas MD Anderson Cancer Center, Houston, TX 77030, USA; civan@mdanderson.org; 3Center for RNA Interference and Non-Coding RNAs, The University of Texas MD Anderson Cancer Center, Houston, TX 77030, USA; 4Department of Translational Molecular Pathology, The University of Texas MD Anderson Cancer Center, Houston, TX 77030, USA; gcalin@mdanderson.org

**Keywords:** ultraconserved region, long non-coding RNA (lncRNA), gastric cancer, epigenetics, lncRNA–lncRNA interaction

## Abstract

**Simple Summary:**

The long non-coding RNA (lncRNA) UC.145, which is conserved in gastric cancer tissue, is presented as a potential novel biomarker for the diagnosis, prognosis, and prediction of gastric cancer. In this study, we analyzed and correlated UC.145 expression with that of DKK1, an inhibitor of the canonical Wnt pathway involved in various cancers. Furthermore, we provide evidence that UC.145 regulates the expression of DKK1 by directly targeting another lncRNA, PRKG1-AS1. Regulation of the three genes was closely associated with the overall survival of patients with gastric cancer.

**Abstract:**

DKK1 inhibits the canonical Wnt signaling pathway that is known to be involved in various cancers. However, whether DKK1 acts as an oncogene or tumor suppressor gene remains controversial. Furthermore, the DKK1-regulating mechanism in gastric cancer has not yet been defined. The aim of this study was to explore whether the ultraconserved region UC.145 regulates epigenetic changes in DKK1 expression in gastric cancer. Microarray analysis revealed that UC.145 exhibited the highest binding affinity to EZH2, a histone methyltransferase. The effects of UC.145 inactivation were assessed in gastric cancer cell lines using siRNA. The results indicated that UC.145 triggers DKK1 methylation via interaction with EZH2 and is involved in the canonical Wnt signaling pathway. Additionally, interaction between UC.145 and another long non-coding RNA adjacent to DKK1, PRKG1-AS1, induced a synergistic effect on Wnt signaling. The regulation of these three genes was closely associated with patient overall survival. Inactivation of UC.145 induced apoptosis and inhibited the growth and migratory, invasive, and colony-forming abilities of gastric cancer cells. The study findings provide insights into Wnt signaling in gastric cancer and support UC.145 as a potential novel predictive biomarker for the disease.

## 1. Introduction

Aberrant activation of the Wnt/β-catenin signaling pathway plays a crucial role in the development and progression of various cancers, including gastric cancer (GC). GC, which has heterogeneous characteristics, including various phenotypes, prognoses, and therapeutic responses, is the fourth leading cause of cancer-related deaths worldwide [1]. Several studies have demonstrated the role of Wnt signaling in the pathogenesis of GC. Dickkopf Wnt signaling pathway inhibitor 1 (DKK1) inhibits Wnt/β-catenin signaling and has been reported in various human diseases, particularly malignant tumors [2,3,4]. Although DKK1 has an antagonistic effect on canonical Wnt signaling as a tumor suppressor, studies report conflicting results regarding DKK1 expression and its role in cancer [3,4,5]. DKK1 expression has been investigated in various cancers; DKK1 is overexpressed in multiple myeloma [6] and breast cancer [7] but weakly expressed in colon [5,8] and cervical cancers [9]. Xiaoting et al. reported that DKK1 expression was downregulated in tissue and serum samples of patients with GC compared to that in healthy controls, whereas its upregulation inhibited the proliferation and invasion of GC cells [10]. Conversely, DKK1 overexpression has also been reported in GC [11]. Hypermethylation of promoter-associated CpG islands of DKK1 has been shown to induce transcriptional inactivation of DKK1, thereby suppressing the inhibition of the Wnt pathway in cancer [12]. Epigenetic regulation of DKK1 mainly occurs in the CpG island of the promoter region, and DKK1 methylation has been reported in some cancers, including gastric cancer, where its expression is silenced [5,7,8,9,11]. However, the upstream mechanisms mediating the epigenetic regulation of DKK1 remain unknown.

Recently, long non-coding RNAs (lncRNAs), which are non-coding transcripts of >200 bp, were reported to epigenetically regulate the Wnt signaling pathway during cancer development [13]. Multiple lncRNAs associated with GC carcinogenesis have been identified, suggesting that lncRNAs may serve as novel biomarkers and targets for the diagnosis and treatment of GC [14,15].

lncRNAs have various biological functions. A new class of non-coding RNAs, termed transcribed ultraconserved regions (T-UCRs), has been associated with both tumor development and suppression [16,17,18,19]. UCRs are completely conserved genomic sequences in humans, rats, and mice [20]. Although T-UCRs have garnered attention as potential biomarkers and therapeutic targets for cancer diagnosis [21], their role in cancer remains largely unknown.

The aim of this study was to investigate the role of DKK1 in GC, lncRNA regulation of DKK1 epigenetic alteration, and the underlying molecular mechanisms. The study findings suggest that non-coding RNAs are involved in the pathogenesis of GC and may serve as potential biomarkers for the disease.

## 2. Materials and Methods

### 2.1. Cell Lines and Cell Culture

GC cell lines AGS (ATCC, Manassas, VA, USA) and MKN74 (Korean Cell Line Bank, Seoul, Korea) were cultured in RPMI 1640 medium (Thermo Fisher Scientific, Waltham, MA, USA) supplemented with 10% fetal bovine serum (Thermo Fisher Scientific) and 1% penicillin/streptomycin (Thermo Fisher Scientific). All cells were incubated under 5% CO_2_ at 37 °C.

### 2.2. Sample Collection and Preparation

A total of 100 samples were obtained from the tissue bank of the Institute of Gastroenterology, Yonsei University College of Medicine (Seoul, Korea), three of which were used for microarray analysis. All patients had undergone surgery for GC, during which cancer and adjacent non-cancer tissues were collected. The tissues were stored at −80 °C. Informed consent was not required because patients had already completed treatment for GC, and no personal identifying information was included with the clinical data and tissue samples. This study was approved by the Institutional Review Board of Yonsei University School of Medicine (IRB number: 4-2013-0024; 7 March 2013).

### 2.3. Antibodies and Reagents

The antibodies used in the study are as follows: DKK1 antibody (Invitrogen, Carlsbad, CA, USA; PA5-23187), Wnt5a antibody (Abcam, Cambridge, UK; ab235966), E-Cadherin antibody (Cell Signaling Technology, Beverly, MA, USA; #3195), N-Cadherin antibody (Cell Signaling Technology; #13116), Vimentin antibody (Cell Signaling Technology; #5741), Histone h3 tri methyl k4 antibody (Abcam; ab8580), Histone h3 tri methyl k9 antibody (Abcam; ab8898), Histone h3 tri methyl k27 antibody (Abcam; ab6002), Histone h3 tri methyl k36 antibody (Abcam; ab9050), Histone h3 antibody (Abcam; ab1791), β-Catenin antibody (Cell Signaling Technology; #8480), c-myc antibody (Cell Signaling Technology; #5605), Cyclin D1 antibody (Cell Signaling Technology; #2978), Snail antibody (Cell Signaling Technology; #3895), PARP antibody (Cell Signaling Technology; #9532), Caspase-9 antibody (Cell Signaling Technology, #9504), Bax antibody (Cell Signaling Technology; #2772), Cytochrome c antibody (Cell Signaling Technology; #4272), Caspase-3 antibody (Cell Signaling Technology; #9662), EZH2 antibody (Abcam; ab186006), and β-actin antibody (Cell Signaling Technology; #47778).

The reagents used in the study included RNAlater (Ambion Inc., Austin, TX, USA); TRIzol reagent (Invitrogen), Superscript II Reverse Transcriptase (Invitrogen), iQ SYBR Green Supermix (Applied Biosystems Inc., Carlsbad, CA, USA), RNAi negative control (Invitrogen), Lipofectamine 2000 reagent (Invitrogen), WST-1 (Promega, Madison, WI, USA), Diff-Quik stain (Dade Behring Inc, Newark, DE, USA), propidium iodide (Sigma-Aldrich, St. Louis, MO, USA), RIPA buffer (Cell Signaling Technology), bovine serum albumin (Affymetrix, Santa Clara, CA, USA), and Magna Chip protein magnetic beads (Merck Millipore, Milan, Italy). The kits used were as follows: CytoSelect™ 96-Well Cell transformation assay kit (Cell Biolabs, San Diego, CA, USA), FITC-Annexin V kit (BD Bioscience, San Jose, CA, USA), DNeasy Blood and Tissue kit (Qiagen, Valencia, CA, USA), EZ DNA Methylation-Gold kit (Zymo Research, Irvine, CA, USA), and High-Sensitivity ChIP kit (Abcam).

### 2.4. Total RNA Extraction and Quantitative Real-Time Reverse Transcription Polymerase Chain Reaction 

Total RNA was extracted from GC cells and tissues using the TRIzol reagent. RNA was quantified using a Multiskan spectrophotometer (Thermo Fisher Scientific). cDNA was synthesized using 2.0 μg RNA with Superscript II in accordance with the manufacturer’s instructions. The expression of UC.145 was determined and analyzed using quantitative real-time polymerase chain reaction (PCR) with a LightCycler 480 instrument and the SYBR Green Supermix. The Ct values of the samples were normalized to U6, GAPDH, and ACTB expression. The 2^−ΔΔCt^ method was used to analyze gene expression. The primer sequences used for qRT-PCR are shown in Appendix A.

### 2.5. Methylation-Specific PCR

Genomic DNA was extracted from AGS and MKN74 cells using the DNeasy Blood and Tissue kit. The EZ DNA methylation kit was used to perform DNA bisulfite conversion. Methylation-specific PCR (MS-PCR) was conducted using specific primers to detect gene methylation status, as listed in Appendix A.

### 2.6. Chromatin Immunoprecipitation Assay

Cells transfected with siControl (siCT) or siUC.145 were crosslinked for 20 min at 25 °C with 1% formaldehyde (Merck Millipore). The crosslinked cells were subjected to ChIP analysis using the High-Sensitivity ChIP kit, according to the manufacturer’s protocol. Real-time PCR analysis was performed using purified DNA samples with the SYBR Green Supermix kit and the primers listed in Appendix A.

### 2.7. Gene Expression Profiling

The Gene Expression Omnibus (GEO) datasets GSE140394, GSE54129, and GSE64951 were obtained and used to identify GC related genes, as well as genes affected by the EZH2 inhibitor. For analysis of RNA interactions with EZH2, RNAInter, PRIdictor, and RPIseq sites were used.

### 2.8. Small Interfering RNA (siRNA) Transfection

Cells were seeded in 6-well plates at a concentration of 3 × 10^5^ cells/well and cultured at 37 °C for 24 h. Transfection with either UC.145 siRNA (50 μM) or RNAi negative control (50 μM, siCT) was performed using Lipofectamine 2000 reagent in Opti-MEM (Invitrogen) media according to the manufacturer’s instructions. All siRNAs were synthesized by Bioneer (Daejeon, Korea) with the following sequences: siUC.145_1 sense: CGG UGC CUG UGA AAU AAC CGA GAU A, anti-sense: U AUC UCG GUU AUU UCA CAG GCA CCG; siUC145_2 sense: GGU GCA CUG UUA AGG UGG AAC CUA A, anti-sense: U UAG GUU CCA CCU UAA CAG UGC ACC; siEZH2_1 sense: CGG AAA CAG GAA CACU GCA UCU UAA, anti-sense: U UAA GAU GCA GUG UUC CUG UUU CCG; siEZH2_2 sense: CAG UAC UUA CCA UUG CUC AUG UUA A, anti-sense: U UAA CAU GAG CAA UGG UAA GUA CUG; siDKK1 sense: CAG CUG UUA GCA GUA AUG CAU UACA, anti-sense: U GUA AUG CAU UAC UGC UAA CAG CUG; siPRKG1-AS1 sense: GCU CCU GAA GUU GUC AUG AGU UUA A, anti-sense: U UAA ACU CAU GAC AAC UUC AGG AGC.

### 2.9. Construction of lncRNA Overexpression Vector

The pc.DNA3.1 (+) expression vector (Addgene, Watertown, MA, USA) was used to amplify UC.145 cDNA using a PCR system (Roche Applied Science, Penzberg, Germany). The cDNA was then inserted into the pcDNA3.1 (+) expression vector UC. 145 _NheI_F (ACCCAAGCTGGCTAGC CGCAGCGAACCCTGCTAAATA). UC. 145_XbaI_R (AAACGGGCCCTCTAGA ATAATTTTTGTTTTAATTGAAAC), PRKG-AS1_ NheI_F (ACCCAAGCTGGCTAGC TGCGAGAAAAATGTGCAAAG), and PRKG1-AS1_ XbaI_R (AAACGGGCCCTCTAGA ACGCGTCTGCCTAATCAAGT) were used as the cloning primers. AGS and MKN74 cells were transfected with 1 μg pcDNA-UC.145 for 24 h using Lipofectamine 2000 (Invitrogen).

### 2.10. Cell Viability Analysis

Cell proliferation was measured in a 96-well plate using the WST-1 assay. GC cells were transfected with 50 nM siUC.145 or siCT, and cell viability was subsequently analyzed using the Multiskan spectrophotometer (Thermo Fisher Scientific) at 450 nm at time points from 0 to 72 h. For rescue experiments, cells were transfected with pcDNA-UC.145 48 h after siRNA transfection, and cell proliferation was subsequently evaluated.

### 2.11. Apoptosis Analysis

AGS and MKN74 cells were transfected with UC.145 siRNA or siCT. After 48 h, cells were centrifuged at 1000× *g* rpm for 2 min at 4 °C. The obtained cell pellet was resuspended in 1× binding buffer (BD Biosciences) and phosphate-buffered saline. Cells were stained with propidium iodide and fluorescein isothiocyanate (FITC) Annexin V using the FITC-Annexin V kit and a FACSverse instrument (BD Biosciences), according to the manufacturer’s instructions. The stained cells were cultured at 37 °C for 15 min and analyzed using a BD FACS Verse II flow cytometer (BD Biosciences). The data were analyzed using FlowJo software version 10.8.1 (Treestar, Ashland, OR, USA).

### 2.12. Invasion and Migration Analysis

After AGS and MKN74 cells were transfected with siUC.145, the invasive ability of cells was assessed using the BD BioCoat Matrigel Invasion Chamber (BD Biosciences) according to the manufacturer’s instructions. Cells that penetrated to the bottom of the insert through the Matrigel were stained with Diff-Quik stain. Invading cells were visualized in five random areas, and total and average cell counts were estimated visually using a BX51 microscope (Olympus, Tokyo, Japan).

For migration analysis, AGS and MKN74 cells (2 × 10^5^ cells) were transfected with siUC.145s or siCT. After incubation for 24 h, wounds were generated using the tip of a P20 pipette. Wound width was measured over time using ImageJ software version 1.8.0 (NIH, Bethesda, MD, USA) with the BX51 microscope. The experiments were performed in triplicate.

### 2.13. Colony Formation Assay

To assess the tumor formation ability of cells, the CytoSelect™ cell transformation assay kit (Cell Biolabs) was employed. To generate the base layer, 1.5 mL of 2X culture media containing 1% agarose was added to each well of a 6-well culture plate. After clotting for 1 h, the transfected cells were mixed with 2X DMEM containing 0.7% agarose, (1:1) added to the base layer, and incubated under 5% CO_2_ at 37 °C for 2–3 weeks. Colonies were observed and imaged daily.

### 2.14. Western Blot Analysis

Cell lysate was obtained using 1X RIPA buffer containing a protease inhibitor (GenDEPOT, Barker, TX, USA) and centrifuged at 2000× *g* rpm for 10 min at 4 °C. Proteins were separated on sodium dodecyl sulfate-polyacrylamide gels and transferred to a polyvinylidene fluoride membrane (Millipore, Darmstadt, Germany). After blocking with 5% bovine serum albumin for 30 min at 25 °C, the membranes were incubated with each primary antibody according to the manufacturer’s instructions and subsequently incubated with an appropriate secondary antibody (GenDEPOT). The membranes were reacted with ECL solution (GenDEPOT), and protein bands were visualized using X-ray film (CP1000; AGFA, Greenville, SC, USA) or ImageQuant LAS 4000 (GE Healthcare, Piscataway, NJ, USA).

### 2.15. RNA Immunoprecipitation

Cells were lysed with IP buffer (Thermo Fisher Scientific) and resuspended in RIP buffer (Abcam) with RNase inhibitor (GenDEPOT) and protease inhibitor (GenDEPOT). Chromatin shearing was performed over 20–30 cycles of shearing, with 15 s of shearing and 30 s of resting on ice for each cycle to maintain cooling conditions in each cycle. Following chromatin shearing, the mixture was centrifuged at 10,000× *g* rpm for 20 min at 4 °C. Subsequently, antibodies were added to the supernatant and the mixture was incubated at 4 °C with constant agitation overnight at 20 rpm. Next, 20 μL MagnaChip protein magnetic beads with affinity for the primary antibody were added and the mixture was incubated at 4 °C for 1 h with constant agitation. After washing twice with RIP buffer, RNA isolation was performed using TRIzol reagent (Invitrogen), according to the manufacturer’s instructions.

### 2.16. Statistical Analysis

The expression of UC.145 in GC was classified as low or high based on the median value. Kaplan–Meier survival curves were constructed and log-rank tests were used to investigate differences in UC.145 expression. The student’s *t*-test was used to compare differences between treatment means. *p*-values < 0.05 and < 0.01 were considered statistically significant. All statistical tests were performed using SPSS (version 18.0; SPSS Inc., Chicago, IL, USA) and Prism software version 8.2.0 (GraphPad Software, San Diego, CA, USA).

## 3. Results

### 3.1. Characteristics of T-UCRs in GC

Microarray analysis was performed using samples collected from three patients with GC to identify differentially expressed T-UCRs in GC. Among 3320 UCR probes investigated in the study, 298 were overexpressed in GC samples, whereas 382 were underexpressed. The remaining probes were not significantly upregulated or downregulated in GC. Considering *p* < 0.05 and fold change > 2.0, 17 T-UCRs were significantly overexpressed in GC (Figure 1a,b). Among 2275 lncRNAs used as positive controls, the expression of 216 lncRNAs was significantly increased in GC tissue, whereas that of 245 lncRNAs was significantly decreased compared to the expression in adjacent normal tissue (*p* < 0.05.) These results were verified in tissue samples from 20 patients with GC using qRT-PCR with primers designed for each T-UCR (Appendix A). 

EZH2 is reportedly involved in the epigenetic regulation of GC [20,22]. A total of 54 candidates were predicted to bind to EZH2 among the non-coding RNAs obtained from the microarray results, scored using the prediction programs RNAInter, PRIdictor, and RPIseq. Finally, 10 T-UCRs and lncRNAs were selected using Venn-diagram-based analysis among 32 candidates for GC. UC.145, which received the highest score among T-UCRs, excluding lncRNAs, was hypothesized to be involved in the development of GC by binding to EZH2 (Figure 1c). Information obtained from the UCSC Genome Browser confirmed that UC.145 is 100% conserved and was predicted to bind to EZH2, although only ChIP-seq data from the ENCODE project were publicly available, and not data from GC cell lines (Figure 1d). UC.145 was overexpressed in five GC cell lines and tissue samples from one patient with GC. Specifically, average UC.145 expression was increased 10-fold in GC cells compared with that in normal gastric cells (GES-1), and particularly high expression was noted in MKN74 cells (Figure 1e). QRT-PCR analysis of 100 paired GC and adjacent normal tissue samples revealed increased UC.145 expression in GC tissue, compared with the adjacent normal tissue (Figure 1f). Moreover, the prediction results suggested that EZH2 and UC.145 function as binding partners to influence the development of GC. RNA immunoprecipitation revealed that UC.145 had a higher binding affinity for EZH2 than for lncRNA HOTAIR (positive control) in AGS and MKN74 cells (Figure 1g). 

EZH2, a subunit of the PRC2 complex, is known for its role in DNA methylation through histone modification by interacting with lncRNAs located primarily in the nuclei [23,24]. Therefore, the location of UC.145 was investigated in GC cells. The presence of UC.145 was confirmed in the nuclei of AGS and MKN74 cells (Figure 1h). Furthermore, transfection with the siRNA vector silenced UC.145 expression (Figure 1i), while that with the overexpression vector led to UC.145 overexpression (Figure 1j) in AGS and MKN74 cells.

### 3.2. UC.145 Regulates In Vitro Proliferation of GC Cells

To confirm whether UC.145 affects the occurrence and development of GC, the effect of UC.145 knockdown was investigated on the proliferation of GC cells treated with siCT or siUC.145s. Compared with cells treated with siCT, AGS, and MKN74, cells treated with siUC.145s showed a significant decrease in the proliferation index (OD value). Meanwhile, cell proliferation inhibition of siUC.145s was partially restored by the overexpression of UC.145 (Figure 2a). In addition, downregulation of UC.145 induced by siRNA inhibited the colony formation ability of GC cells (Figure 2b). These findings indicated that UC.145 could regulate the proliferation of GC cells. Furthermore, the flow cytometry results indicated increased apoptosis and cell cycle arrest in cells transfected with siUC.145s (Figure 2c,d). Additional experiments revealed that pro- and anti-apoptotic genes were regulated by UC.145 (Appendix A).

Additionally, UC.145 knockdown reduced the migratory and invasive abilities of GC cells (Figure 2e,f). Notably, transfection with siUC.145s regulated the expression of mesenchymal markers (Appendix A). Meanwhile, these functions were restored following treatment with pcDNA-UC.145.

### 3.3. DKK1 Is a Target of UC.145 in GC

To investigate the pathway through which UC.145 is involved in the progression of GC, gene expression array analysis was performed following downregulation of UC.145 in AGS cells. The Nanostring nCounter Pancancer pathway panel was used to detect the effects of cancer-related gene expression (Figure 3a, Appendix A). Among 770 genes identified, five were upregulated and five were downregulated following UC.145 silencing. The expression of these 10 genes was verified using qRT-PCR. Among them, the expression of DKK1 was significantly upregulated after treatment with siUC.145s (Appendix A), suggesting that DKK1 is a target of UC.145. Therefore, to determine the association between UC.145, DKK1 [25,26,27], and EZH2 in GC, Venn-diagram-based analysis was performed using GEO datasets GSE140394 [28], GSE54129 [29], and GSE64951 [30]. The results of the Venn-diagram-based analysis showed consistent upregulation of five genes, suggesting that inhibition of UC.145 and EZH2 was associated with the development of GC (Figure 3b). Furthermore, following treatment of cells with siUC.145s, mRNA and protein levels of DKK1 were increased, as determined using qRT-PCR (Figure 3c) and Western blotting (Figure 3d). The results confirmed that DKK1 is regulated by UC.145. Collectively, these results demonstrated that downregulation of UC.145 increases DKK1 expression, indicating that DKK1 is a target of UC.145 in the development of GC.

Expression of the remaining four genes identified in the Venn-diagram-based analysis was also confirmed to be regulated by UC.145 (Appendix A). Although these four genes—LAMA5 [27], IGF1 [31], CDKN1A [32], and CREB5 [33]—reportedly participate in Wnt signaling, DKK1 was selected for further analysis since it was most significantly regulated by UC.145. 

DKK1 is methylated in several cancers, including glioblastoma [34], renal cell carcinoma [5], hepatic fibrosis [25], and breast cancer [7]. Therefore, MS-PCR was performed to determine whether DKK1 is methylated in GC and whether this process is controlled by UC.145 binding to EZH2. The results indicated that DKK1 was methylated in AGS and MKN74 cells. Moreover, treatment of GC cells with UC.145-targeting siRNA verified that UC.145 is involved in this methylation process (Figure 3e). Specifically, UC.145 was confirmed to regulate DKK1 methylation. These results suggested that UC.145 might participate in this epigenetic control by binding to EZH2, which required further verification.

DKK1 expression and its association with UC.145 was further investigated in GC, since relatively little is known regarding the role DKK1 plays in GC. qRT-PCR analysis revealed no difference in DKK1 expression in the 100 tissue pairs collected from patients with GC (Appendix A). Subsequently, the results were divided into high or low UC-145 expression groups based on the median value of UC.145 expression. Notably, DKK1 expression was significantly lower (*p* < 0.0001) in the high-UC.145 group than in the low-UC.145 group (Figure 3f). Collectively, these results demonstrated a negative correlation between UC.145 and DKK1 (Figure 3g).

To investigate this relationship more closely, DKK1 protein levels and UC.145 expression were measured in 10 randomly selected GC tissues using Western blotting and qRT-PCR (Figure 3h), which re-verified their negative correlation (Figure 3i). Further, changes in UC.145 expression were investigated after suppression of DKK1 using siDKK1. However, UC.145 expression was not affected by treatment with siDDK1 (Appendix A), suggesting that UC.145 may function upstream of DKK1. Next, the effect of DKK1 and UC.145 expression on the proliferation of AGS and MKN74 cells was explored using siUC.145s and siDKK1. Knockdown of DKK1 partially restored the reduced proliferation induced by siUC.145s (Figure 3j). These findings suggested that UC.145 and DKK1 counteract the proliferative potential of GC cells.

### 3.4. UC.145 Regulates DKK1 by Interacting with EZH2

The EZH2-binding site of DKK1 was investigated using the ChIP-seq dataset from the publicly available ENCODE project. The promoter region of DKK1, a target of UC.145, was found to contain the EZH2-binding site enhancer (Figure 4a). RNA immunoprecipitation verified that the promoter region of DKK1 bound to EZH2 (Figure 4b), indicating that DKK1 is regulated by EZH2. To further confirm these findings, EZH2-targeting siRNA was synthesized, the efficacy of which was verified at the mRNA (Figure 4c) and protein (Figure 4d) levels. DKK1 was regulated by EZH2 at the mRNA (Figure 4e) and protein (Figure 4f) levels and methylated (Figure 4g) in AGS and MKN74 cells. In addition, downregulation of UC.145 decreased EZH2 protein expression in AGS and MKN74 cells (Figure 4h), whereas no changes were observed in mRNA expression (Appendix A). Moreover, considering that DNA methylation is related to histone modification [35], ChIP-PCR was performed to assess changes in histone modification markers. The results indicated that H3K27me3 signaling was considerably reduced via histone modification by regulating UC.145 (Figure 4i). Furthermore, inhibition of UC.145 expression decreased the expression of H3K27me3-modified global markers (Figure 4j). Finally, to demonstrate the association between EZH2 and DKK1 regulation by UC.145, cells were treated with an EZH2 inhibitor, GSK126, which resulted in DKK1 upregulation (Figure 4k).

### 3.5. Novel lncRNA PRKG1-AS1, Which Is Controlled by UC.145, Regulates DKK1

The lncRNA PRKG1-AS1 was identified near DKK1 as an antisense RNA. Although the function of this lncRNA was unknown, it was hypothesized that PRKG1-AS1 might regulate DKK1 and interact with UC.145. To test this hypothesis, PRKG1-AS1 expression in GC cells and normal gastric cells (GES-1) was compared; however, no difference in expression was identified (Appendix A). In addition, PRKG1-AS1 expression did not differ in the 100 pairs of tissues collected from patients with GC (Appendix A).

Notably, the inhibition of UC.145 using siRNA increased PRKG1-AS1 expression in AGS and MKN74 cells, demonstrating a relationship between the two lncRNAs (Figure 5a). Meanwhile, the high-UC.145 group demonstrated lower PRKG1-AS1 expression than the low-UC.145 group (Figure 5b), suggesting an inverse correlation (Figure 5c). Additionally, overexpression of PRKG1-AS1 obtained using an overexpression vector resulted in downregulation of UC.145 (Figure 5d).

Next, the effect of intentional overexpression of PRKG1-AS1 on DKK1 regulation was investigated, as well as whether UC.145 was involved in this process. Overexpression of PRKG1-AS1 increased DKK1 mRNA (Figure 5e) and protein (Figure 5f) levels in both AGS and MKN74 cells. The DKK1 regulatory mechanism also appeared to participate in methylation control (Figure 5g), leading to decreased EZH2 protein levels (Figure 5h). Moreover, UC.145 was found to be involved in these processes. Specifically, overexpression of PRKG1-AS1 decreased the binding affinity between EZH2 and UC.145, similar to the interaction between lncRNA and miRNA (Figure 5i).

Subsequently, the effect of UC.145 and PRKG1-AS1 interaction on DKK1 expression and Wnt signaling was investigated in AGS and MKN74 cells using Western blotting. Suppression of UC.145 increased DKK1 expression but decreased Wnt5a and β-catenin expression. Notably, regulation of PRKG1-AS1 expression restored the changes in protein expression caused by UC.145 (Figure 5j). 

Survival analysis of 100 patients with GC revealed a significant association between overall survival, UC.145 (log-rank *p*-value = 0.0116), its target DKK1, and its interacting molecule PRKG1-AS1 (log-rank *p*-value = 0.0117) (Figure 5k). Survival analysis according to cancer stage was also performed (Appendix A), revealing that stage III was most associated with UC.145, DKK1, and PRKG1-AS1. These findings supported that UC.145 and DKK1 are involved in the development of GC, particularly via regulation of Wnt signaling.

## 4. Discussion

In this study, microarray analysis for T-UCRs was performed to determine the involvement of lncRNAs in the epigenetic regulation of the Wnt signaling pathway, an important carcinogenic mechanism in GC [36,37]. Various bioinformatic and laboratory methods were employed to identify the relationship with EZH2, a major epigenetic regulator. In particular, one of the aims of the study was to determine why the expression of DKK1, a potent inhibitor of the Wnt signaling pathway in many cancers, has been shown to have conflicting effects in GC.

EZH2 expression, which is controlled by multiple lncRNAs, is upregulated in various cancers. Moreover, EZH2 inhibits tumor suppressor genes via methylation [22]. Therefore, the UCR that targeted EZH2 was selected in this study. We previously reported that an lncRNA G-rich motif is important for the binding affinity of EZH2 [13]. Although the precise mechanism underlying the binding of UC.145 to EZH2 was not characterized in the previous study, the results suggest that the G-rich motif of UC.145 interacts with the N-terminal helix of EZH2 and is involved in the occurrence of GC.

Generally, lncRNAs affect cancer development and metastasis via cis-acting elements that occur on the same chromosome, or trans-acting elements on different chromosomes [38]. Because genes adjacent to UC.145 were not detected in the current study, it was hypothesized that UC.145 functions by trans-acting elements. To unambiguously identify the target of UC.145, candidates were selected among cancer related genes using gene expression array analysis, ultimately revealing DKK1 as a target of UC.145. In a previous study, we found that lncRNA regulates EZH2 expression and epigenetically regulates methylation in the CpG island present in the promoter region of tumor suppressor genes, and this hypothesis was introduced in this study [13].

The study findings indicated that only the expression of UC.145, not DKK1, was significantly increased in GC tissues, and UC.145 oncogenic activity was further confirmed by inducing cell proliferation and colony formation, reducing apoptosis, and arresting the cell cycle in GC cells. UC.145 increased EZH2 expression, ultimately inducing promoter methylation of DKK1 and suggesting a negative correlation between DKK1 and UC.145 in GC. Additionally, PRKG1-AS1 antagonistically interfered with the interaction between UC.145 and EZH2. The results suggest that PRKG1-AS1, EZH2, and DKK1 are reciprocally involved in UC.145-driven changes in the Wnt signaling pathway in GC, thus reflecting the diversity and complexity of the development process of GC.

Despite recent advances, biomarkers have limited clinical use in cancer treatment. This study helps address some of these limitations. Involvement of a UCR, which is conserved in various species including humans, in the proliferation and apoptosis of GC cells suggests a novel mechanism in GC. DKK1 expression regulated by UC.145 inhibited the proliferation of GC cells. Furthermore, high DKK1 expression was associated with poor prognosis in GC, whereas re-evaluation of survival analysis focusing on UC.145 associated DKK1 expression displayed the opposite trend (Appendix A). This finding raises the expectation that UC.145 and DKK1 have potential for therapeutic and diagnostic purposes.

Re-evaluation of genes that are non-specifically expressed in cancer, such as PRKG1-AS1, may help further elucidate the heterogeneous pathogenesis of GC. Although many biomarkers for GC have been reported in previous studies, their clinical applicability is remarkably low due to their lack of specificity [10]. However, the results of the current study reveal that PRKG1-AS1, which is non-GC-specific, is regulated by UC.145 and mediates DKK1 expression through EZH2 regulation, suggesting that it may be involved in GC development.

These findings also help explain why contradictory results have been previously reported for DKK1 expression in GC. DKK1 may not primarily regulate the Wnt signaling pathway in GC. Instead, UC.145, reported here for the first time, might regulate this pathway. In addition, regulation of PRKG1-AS1 expression by UC.145 was associated with a favorable prognosis during survival analysis (Appendix A). These results suggest that UC.145 and PRKG1-AS1, lncRNAs that regulate DKK1, may be considered when GC prognosis is based on DKK1 or as potential diagnostic markers. Furthermore, the study suggests the possibility of a new biomarker for GC and demonstrates a novel approach in cancer research [15,39]. Nevertheless, further studies are warranted to elucidate the relationship among the three genes using molecular approaches and assess their clinical applicability. Notably, because reports of other regulatory mechanisms with known coding genes remain inconsistent, the study findings support that non-coding RNAs should be considered.

## 5. Conclusions

In this study, lncRNA UC.145 expression was analyzed and correlated with DKK1 expression in GC. The study provides evidence that UC.145 regulates the expression of DKK1 by directly targeting another lncRNA, PRKG1-AS1. Additionally, the results support that UC.145 may provide a potential diagnostic and predictive biomarker for GC to help further understand the pathogenesis of GC and address the problem of the disease’s heterogeneity. Considering the inconsistent results reported for many GC-associated genes, non-coding RNA may be the key to elucidating the molecular mechanisms underlying GC. The study findings provide a basis for the development of novel lncRNAs as potential biomarkers.

## Figures and Tables

**Figure 1 cancers-14-02369-f001:**
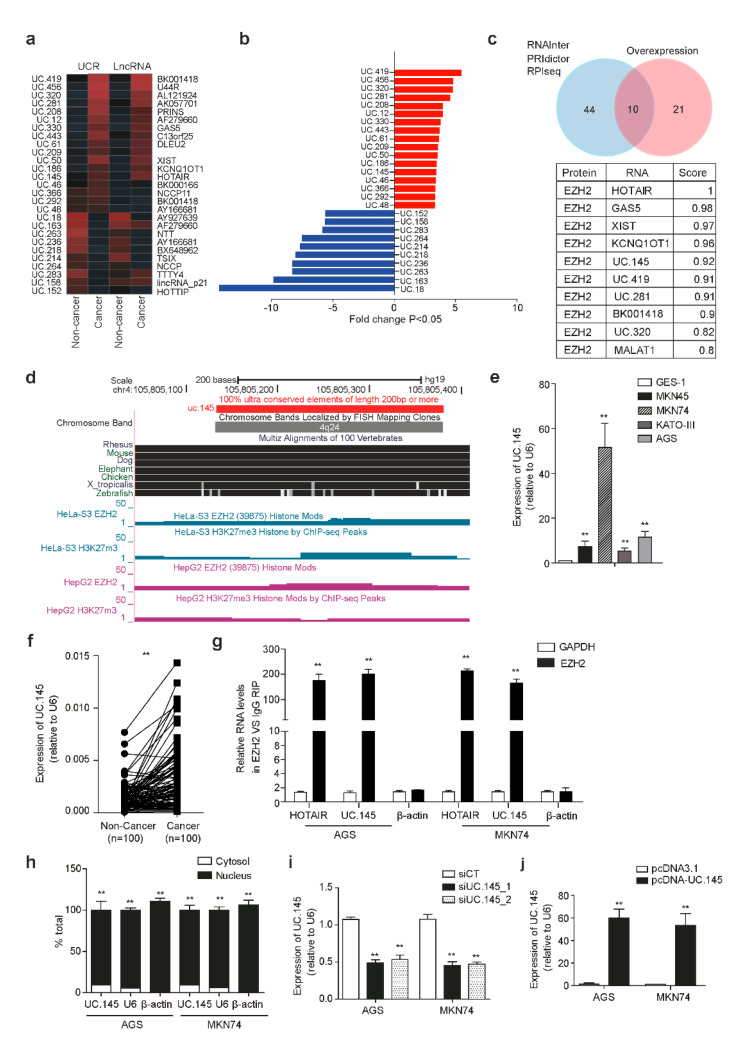
Characteristics and expression of UC.145 in gastric cancer (GC)**.** (**a**) Transcribed ultraconserved region (T-UCR) expression was altered in GC tissues compared with that in adjacent healthy tissues. Rows represent individual T-UCRs. Columns represent GC samples. Heat maps of long non-coding RNA (lncRNA) and T-UCR are presented using hierarchical cluster analysis (FC > 2.0, *p* < 0.05); (**b**) Expression of T-UCR; (**c**) Venn-diagram-based analysis of ten non-coding RNAs, including UC.145, using targeted EZH2 (blue circle) predicted using RNAInter, PRIdictor, and RPIseq, and upregulated non-coding RNAs obtained by Agilent human chip analysis (red circle). Interaction scores are also displayed; (**d**) Genetic information of UC.145 obtained from the UCSC Genome Browser database; (**e**) UC.145 expression measured in five GC cell lines and one normal human gastric epithelial cell line (GES-1); (**f**) UC.145 expression measured using qRT-PCR for 100 paired GC and normal tissue samples; (**g**) RIP analysis performed with the anti-EZH2 antibody using AGS and MKN74 cell lysates. UC.145 expression in cell lysates and immunoprecipitates measured using qRT-PCR. LncRNA HOTAIR served as the positive control; (**h**) UC.145 expression in nuclear and cytoplasmic fractions of AGS and MKN74 cells measured using qRT-PCR; (**i**) Knockdown effect of siRNA on UC.145 expression in AGS and MKN74 cells measured using qRT-PCR; (**j**) Increased expression of UC.145 upon transfection with overexpression vector in AGS and MKN74 cells measured using qRT-PCR. Data are shown as mean ± SD of three independent experiments (** *p* ≤ 0.01).

**Figure 2 cancers-14-02369-f002:**
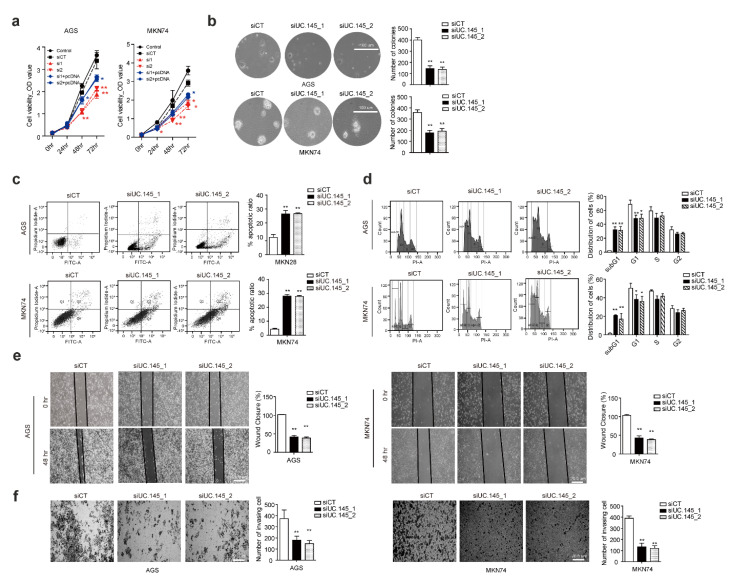
UC.145 regulates the phenotype of gastric cancer (GC) cells. (**a**) Cell viability was determined using the WST-1 assay in AGS and MKN74 cells transfected with siControl (siCT), siUC.145 (si1, si2), or siUC.145 + pcDNA-UC.145 (si1 + pcDNA, si2 + pcDNA). Growth curves were compared for transfected cells. (**b**) The colony formation assay was performed after AGS and MKN74 cells were transfected with siUC.145. (**c**) Apoptosis was induced by suppressing UC.145 expression using siRNA, measured by flow cytometry using PI/Annexin V staining. Bar graphs show the percentage of apoptotic cells in each population. (**d**) To analyze cell cycle changes, AGS and MKN74 cells were transfected with siRNA and analyzed by flow cytometry using PI staining. Bar graphs show the percentage of cells in the lower G0, G1, S, and G2 stages in siRNA-transfected cell populations. Wound healing assays (**e**) and invasion assays (**f**) were performed with AGS and MKN74 cells after UC.145 knockdown by siUC.145. Bar graphs represent the frequency of wound closures, number of invading cells, and colonies formed. Data are shown as mean ± SD of three independent experiments (* *p* ≤ 0.05; ** *p* ≤ 0.01).

**Figure 3 cancers-14-02369-f003:**
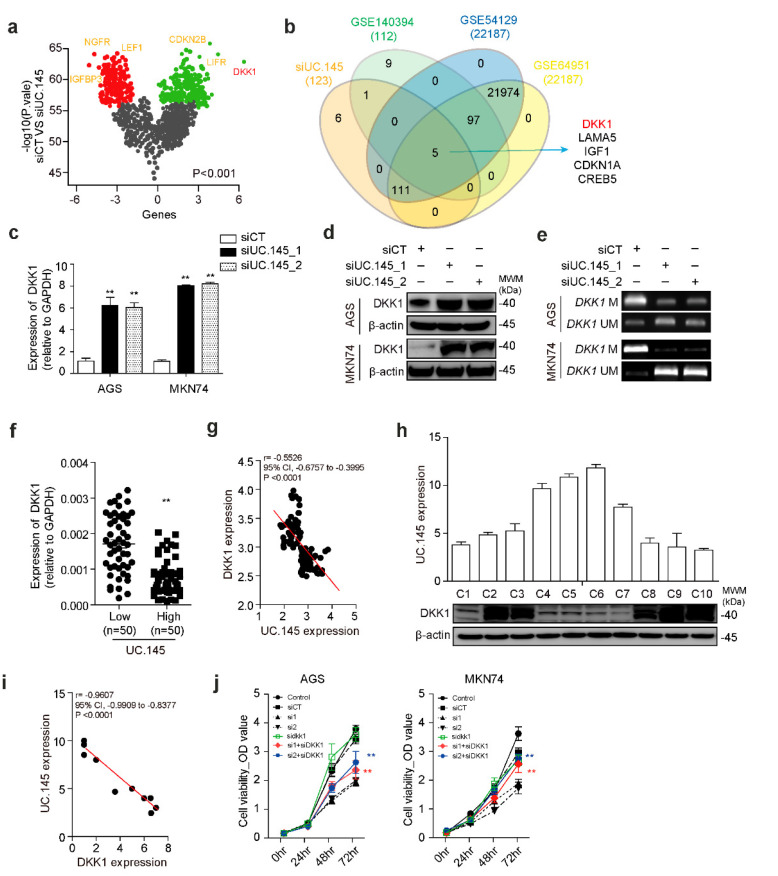
UC.145 modulates the methylation of DKK1 in gastric cancer (GC). (**a**) Changes in expression of cancer related genes following siUC.145 treatment are shown compared to treatment with siControl (siCT). UC.145 is upregulated (green) or downregulated (red) under depleted conditions. The three genes with the most significantly altered expression are presented; (**b**) Expression of five genes simultaneously altered by siUC.145, EZH2 inhibitor, and GC related genes (GSE140394, genes involved in EZH2 suppression; GSE54129 and GSE6495, genes involved in GC); (**c**) DKK1 expression validated in AGS and MKN74 cells using qRT-PCR; (**d**) DKK1 protein changes in AGS and MKN74 cells transfected with siCT or siUC.145 confirmed by Western blotting; (**e**) Methylation changes detected using MS-PCR; (**f**) DKK1 expression in GC tissues according to classification based on the median expression of UC.145 using qRT-PCR; (**g**) Correlation between DKK1 mRNA level and UC.145 expression in GC samples quantified using the densitometry method with Prism; (**h**) UC.145 expression in 10 randomly selected GC samples using qRT-PCR. Western blotting results performed on the same tissues; (**i**) Correlation between UC.145 expression and DKK1 protein level in GC tissue quantified using density measurement with Prism; (**j**) Effect of DKK1 expression on cell proliferation in AGS and MKN74 cells in which the expression of UC.145 was suppressed (siControl (siCT), siUC.145 (si1, si2), or siDKK1). Data are shown as mean ± SD of three independent experiments (** *p* ≤ 0.01).

**Figure 4 cancers-14-02369-f004:**
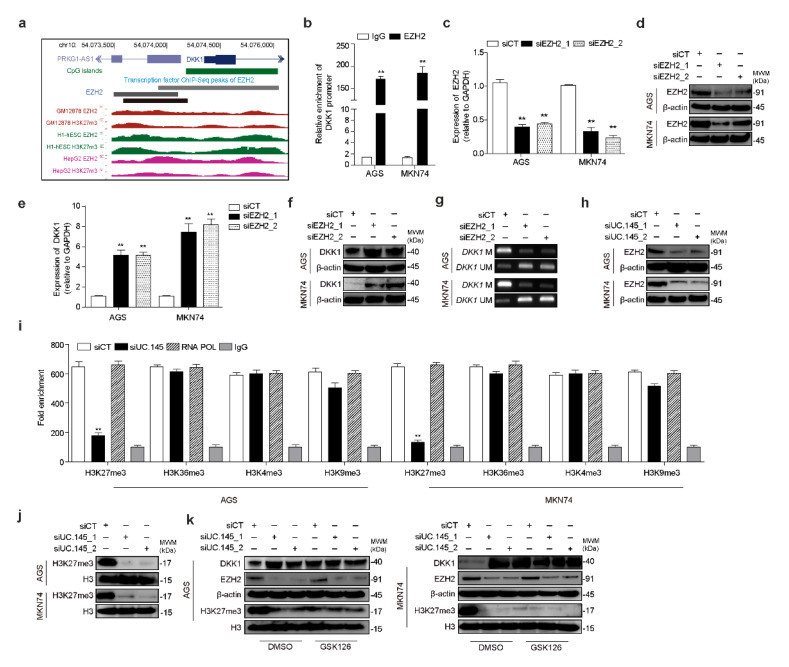
UC.145 epigenetically regulates DKK1 involved in the canonical Wnt signaling pathway. (**a**) EZH2 ChIP-seq signal identified at the DKK1 genomic locus using the CpG island region, EZH2 ChIP-seq peak, and H3K27me3 signal using UCSC Genome Browser; (**b**) RNA immunoprecipitation results determining that EZH2 binds to the DKK1 promoter region, confirmed using qRT-PCR. EZH2 targeting siRNA suppressed DKK1 expression at the (**c**) mRNA and (**d**) protein levels. Altered DKK1 expression was confirmed using (**e**) qRT-PCR and (**f**) Western blotting. (**g**) DKK1 methylation was confirmed using MS-PCR. (**h**) Western blotting confirmed that altered EZH2 abundance was caused by inhibition of UC.145 expression; (**i**) ChIP-qPCR analysis of histone modification markers in siControl- or siUC.145-transfected cells; (**j**) Western blotting confirmed changes in histone modification markers after treatment with siUC.145s. (**k**) Changes in DKK1, EZH2, and H3K27me3 expression regulated by UC.145 were confirmed by Western blotting following treatment with EZH2 inhibitor GSK126. The results in (**b**–**k**) were obtained in AGS and MKN74 GC cells. Data are shown as mean ± SD of three independent experiments (** *p* ≤ 0.01).

**Figure 5 cancers-14-02369-f005:**
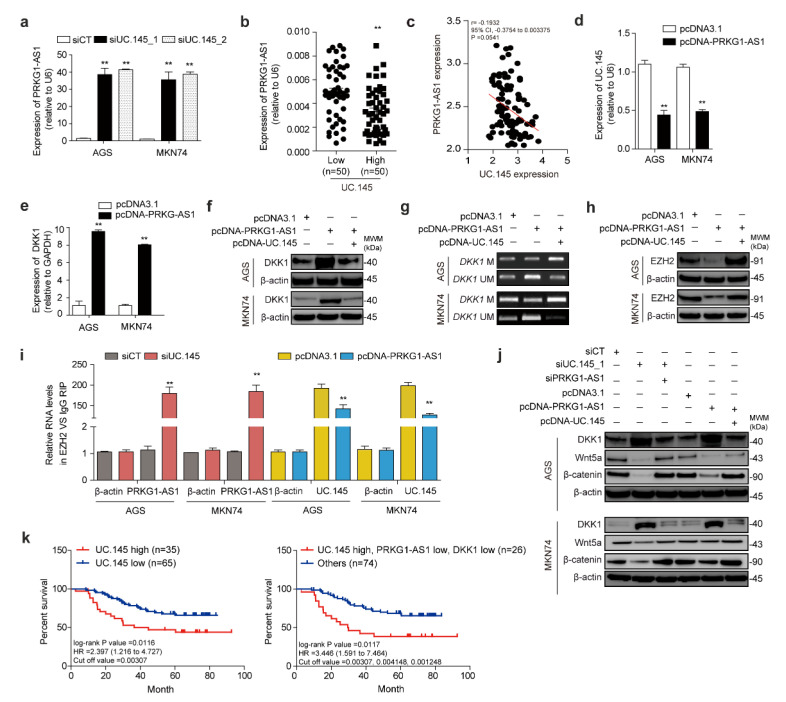
lncRNA PRKG1-AS1 regulated by UC.145 affects DKK1 expression. (**a**) PRKG1-AS1 expression was confirmed under UC.145 expression using qRT-PCR; (**b**) PRKG1-AS1 expression in GC samples was confirmed according to classification based on the median expression of UC.145 using qRT-PCR; (**c**) Correlation between UC.145 and PRKG-AS1 analyzed using Prism software; (**d**) Change in UC.145 and (**e**) DKK1 expression confirmed under PRKG1-AS1 overexpression using qRT-PCR. PRKG-AS1 overexpression caused changes in (**f**) DKK1 protein level (detected using Western blotting), (**g**) methylation using MS-PCR, and (**h**) EZH2 protein level (using Western blotting); (**i**) Silencing UC.145 via siRNA or ectopic overexpression of PRKG1-AS1 confirmed EZH2 binding using qRT-PCR through the RIP assay; (**j**) Western blotting of DKK1, Wnt ligand Wnt5a, and β-catenin components of Wnt/β-catenin following the modulation of UC.145 and PRKG1-AS1 expression. (**k**) Kaplan–Meier estimates of the overall survival rate of patients with high or low UC.145 expression; log-rank *p* = 0.0116 (left panel). The overall survival rate of patients with high expression of UC.145 and low expression of DKK1 and PRKG1-AS1. Log-rank *p* = 0.0117 (right panel). Data are shown as mean ± SD of three independent experiments (** *p* ≤ 0.01).

## Data Availability

All data generated or analyzed during this study are included in this published article and its Appendix A Files.

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
