# Peer review of "lncRNAs UC.145 and PRKG1-AS1 Determine the Functional Output of DKK1 in Regulating the Wnt Signaling Pathway in Gastric Cancer"

_cancers, 2022, doi:10.3390/cancers14102369_

Round 1

Reviewer 1 Report

manuscript  is  corrected as  recommended  now can  be  published

Author Response

Response letter attached.

Reviewer 2 Report

Overall, the revised manuscript is much improved.  However, some of the original reviewer comments were not adequately addressed. These omissions/mistakes need to be corrected.

General comment 3: 

Reviewer comment: Furthermore, data presented in Fig 5j do not well match the text (Lines 421-423). With respect to the combinations (si+si or OE+OE), where are the single construct controls?

Author response: We have corrected the contents of lines 477-481 as follows: “Subsequently, the effect of UC.145 and PRKG1-AS1 interaction on DKK1 expression and Wnt signaling was investigated in AGS and MKN74 cells using western blotting. Suppression of UC.145 increased DKK1 expression, but decreased Wnt5a, MMP7, and β-catenin expression. Notably, regulation of PRKG1-AS1 expression restored changes in gene expression caused by UC.145 (Fig. 5j).”

Reviewer response: The results as written, “Suppression of UC.145 increased DKK1 expression, but decreased Wnt5a, MMP7, and β-catenin expression. Notably, regulation of PRKG1-AS1 expression restored changes in gene expression caused by UC.145” are not actually shown in Figure 5j. According to the -/+ labels above the Western blot images, there is no siUC.145_1 in this figure.  The authors either need to include the siUC.145_1 sample in the figure or correct the labeling if it is erroneous. In addition, the authors refer to “gene” expression, but Westerns show only protein expression.

Specific comment 8: 

Reviewer comment: Line 264 is confusing and should be edited for clarity

Author response:  We appreciate the reviewer’s attention to detail. We have modified this sentence to clarify our meaning. We apologize for our error in communicating the results. I think the revised sentence will help you understand our intended meaning: “EZH2 is reportedly involved in the epigenetic regulation of GC [20, 22]. A total of 54 candidates were predicted to bind to EZH2 among the noncoding RNAs obtained from the microarray results, scored using the prediction programs RNAInter, PRIdictor, and RPIseq.”

Reviewer response: This response has nothing to do with Line 264 in the original manuscript, which are Lines 312-314 in the revised manuscript, “Meanwhile, cell proliferation was inhibited after transfection with the UC.145 overexpression vector at 48 h (Fig. 2a)”. My guess is that the authors mean that cell proliferation of siUC.145 cells was partially restored by overexpression of UC.145. However, as written, the result does not match the data presented in the figure, nor does it make sense.

Author Response

Response letter attached. 

This manuscript is a resubmission of an earlier submission. The following is a list of the peer review reports and author responses from that submission.

Round 1

Reviewer 1 Report

Abstract  should  be  more  clear  and  concise  with  the  focus  of these  three interacting  genes. In  the Abstract  conclusion  define exact correlation  of these three interacting  genes

In the introduction and the  discussion  elaborate  more on methylation  process of these interacting  genes

Figure 3h Correlation  of the UC.145 expression and DKK1  should  be  supported by the   mutilation of DKK1

Fig  4 , and Fig 5 also  methylation of the DKK1 needs to be shown

Author Response

Response to Reviewer 1

Comments and Suggestions for Authors

Abstract  should  be  more  clear  and  concise  with  the  focus  of these  three interacting  genes. In  the Abstract  conclusion  define exact correlation  of these three interacting  genes

  • Thank you for this suggestion. The abstract has been edited to describe the results of the study more clearly.

In the introduction and the  discussion  elaborate  more on methylation  process of these interacting  genes

We addressed the above comment in Introduction on line 57 “Epigenetic regulation of DKK1 mainly occurs in the CpG island region of the promoter region, and it has been reported that DKK1 methylation has been found in some cancers including gastric cancer, and its expression is silenced. [5,7,8-9,11].,”

We addressed the above comment in Discussion on line 508 “In a previous study, we found that lncRNA regulates the expression of EZH2 and epigenetically regulates methylation in the CpG island present in the promoter region of tumor suppressor genes, and this hypothesis was introduced in this study [13].”

Figure 3h Correlation  of the UC.145 expression and DKK1  should  be  supported by the   mutilation of DKK1

  • Thank you for providing this helpful comment. Fig. 3h presents data confirming that UC.145 expression affects the DKK1 protein level in randomly selected tissues from patients with gastric cancer. These results demonstrate the conflicting relationship between UC.145 and DKK1 The different levels of expression in each tissue sample are likely due to the heterogeneity of gastric cancer. To determine the relationship between these two genes, we knocked down DKK1 and assessed the change in UC.145 expression. The results indicated that UC.145 expression was unchanged by regulating DKK1 expression (Fig. S7), suggesting that UC.145 is involved in gastric cancer development upstream of DKK1 in the signaling pathway. These results are described in lines 397-400 in the results section.

Figure S7. DKK1 affects UC.145 at the transcriptional level in GC cells. Relative expression of UC.145 in AGS and MKN74 cells, measured using qRT-PCT.

Fig  4 , and Fig 5 also  methylation of the DKK1 needs to be shown

  • Thank you for providing this suggestion. We missed the experiment that would confirm DKK1 methylation by inhibiting EZH2 expression. Based on the reviewer's comments, we have performed MS-PCR and confirmed that the regulation of EZH2 expression changes the level of DKK1 methylation. The results demonstrated that EZH2 regulates DKK1 methylation in gastric cancer cells. We have presented these results in Fig. 4g and on line 414 in the results section.

Figure 4. (g) UC.145 epigenetically regulates DKK1 involved in the canonical Wnt signaling pathway. DKKI methylation was confirmed using MS-PCR.

Reviewer 2 Report

In this study, Yoon et al. determined a role for the long non-coding RNA (lncRNA) UC.145 in epigenetic regulation of Dickkopf Wnt signaling pathway inhibitor 1 (DKK1) in gastric cancer (GC), using a combination of patient-derived tumors and GC cell lines. UC.145 was identified by microarray analysis to have increased expression in GC compared to normal adjacent tissue. The inactivation of UC.145 by siRNA in two gastric cancer cell lines (MKN28, MKN74) resulted in reduced proliferation, colony formation and migration/invasion, and increased apoptosis. The authors show that regulation of DKK1 by UC.145 is mediated by the methyltransferase Enhancer of zeste homologue 2 (EZH2).

Overall, the manuscript is well written, and the study seems to be well performed, very mechanistic.  However, some of the figures are either not inserted correctly or absent altogether, making it difficult to evaluate all the data. In addition, there a number of issues to address before it is suitable for publication.

General comments: 

1. Figure 2 is upside-down, and the Supplementary Figures were not included. Thus, these data have not been critically reviewed. Overall, there is a lack of sufficient proofreading of the figure content in this manuscript.

2. It is important to have multiple cells lines to rule out cell line specificity. However, the GC cell line MKN28 has been reported to be problematic [Expasy’s Cellosaurus] since it has been shown to be derived from MKN74 (ie. cross-contaminated). It is notable that aside from Fig 1e data, MKN28 and MKN74 behave/respond very similarly in most assays. The authors need to authenticate these cell lines by STR profile analysis and provide the results to the reviewers.

3. The relationship between UC.145, PKRG1-AS1, and DKK1 is hard to follow/not clear. Data showing that UC.145 and PRKG1-AS1 are oppositely expressed in GC tissue are not strong (Fig 5b-c) and do not appear to be statistically significant. Furthermore, data presented in Fig 5j do not well match the text (Lines 421-423). With respect to the combinations (si+si or OE+OE), where are the single construct controls? This figure/section of the manuscript (also Lines 32-33 of the Abstract) need to be combed through and carefully edited for clarity.

Specific comments: 

1. Line 45: According to the WHO, GC is the fourth leading cause of cancer-related deaths (not third).

2. Source of cell lines: None of the cell lines listed in the Methods sections are on ATCC's website, as indicated on Line 77. Furthermore, all cell lines used in the study and their appropriate culture conditions should be included here.

3. All siRNA sequences used (ie. DKK1, EZH2) should be included in the Methods.

4. Antibody information including manufacturer/source should be added to the Methods.

5. Fig 1b and 1d are too small and cannot be read at their current sizes

6. Since the GC + adjacent normal tissue are matched pairs, Fig 1f graph would be better presented as paired data points (in the format of “before-after”) with a paired t-test used for statistical analysis

7. What is the meaning of the extra groups of columns in Fig 1g-h graphs with no labels or mention in the text?

8. Line 264 is confusing and should be edited for clarity

9. Fig 3j results cannot be properly interpreted without a siDKK1 alone control. If KD of DKK1 increases proliferation on its own (very possible since it is a Wnt inhibitor), then the combination of siDKK1 and siUC145 is additive rather than “partially restored”

10. Include Westerns to verify KD of EZH2 by siRNAs

11. Fig 4i is missing DMSO/GSK126 labels

Author Response

Response to Reviewer 2

Comments and Suggestions for Authors

In this study, Yoon et al. determined a role for the long non-coding RNA (lncRNA) UC.145 in epigenetic regulation of Dickkopf Wnt signaling pathway inhibitor 1 (DKK1) in gastric cancer (GC), using a combination of patient-derived tumors and GC cell lines. UC.145 was identified by microarray analysis to have increased expression in GC compared to normal adjacent tissue. The inactivation of UC.145 by siRNA in two gastric cancer cell lines (MKN28, MKN74) resulted in reduced proliferation, colony formation and migration/invasion, and increased apoptosis. The authors show that regulation of DKK1 by UC.145 is mediated by the methyltransferase Enhancer of zeste homologue 2 (EZH2).

Overall, the manuscript is well written, and the study seems to be well performed, very mechanistic.  However, some of the figures are either not inserted correctly or absent altogether, making it difficult to evaluate all the data. In addition, there a number of issues to address before it is suitable for publication.

General comments: 

  1. Figure 2 is upside-down, and the Supplementary Figures were not included. Thus, these data have not been critically reviewed. Overall, there is a lack of sufficient proofreading of the figure content in this manuscript.
  • We apologize for this mistake and the inconvenience it caused. We did not notice that Figure 2 was upside down when we submitted the manuscript. A compatibility error must have occurred in Illustrator. The orientation of the figure has been corrected in the re-submitted manuscript. Additionally, the supplementary figures have been submitted with the revised manuscript.
  1. It is important to have multiple cells lines to rule out cell line specificity. However, the GC cell line MKN28 has been reported to be problematic [Expasy’s Cellosaurus] since it has been shown to be derived from MKN74 (ie. cross-contaminated). It is notable that aside from Fig 1e data, MKN28 and MKN74 behave/respond very similarly in most assays. The authors need to authenticate these cell lines by STR profile analysis and provide the results to the reviewers.
  • When selecting a cell line at the beginning of the study in 2012, we were not aware of cross contamination between MKN28 and MKN74. We should have changed the cell line in future research, and we regret that we performed our research without recognizing this information. Admitting this mistake, we reran all the experiments using the AGS cell line and presented these results in the revised manuscript. We apologize once again for not being aware of the problem with the cell line in our original study, and we hope you will see that we have resolved this issue in the resubmitted manuscript.
  1. The relationship between UC.145, PKRG1-AS1, and DKK1 is hard to follow/not clear. Data showing that UC.145 and PRKG1-AS1 are oppositely expressed in GC tissue are not strong (Fig 5b-c) and do not appear to be statistically significant. Furthermore, data presented in Fig 5j do not well match the text (Lines 421-423). With respect to the combinations (si+si or OE+OE), where are the single construct controls? This figure/section of the manuscript (also Lines 32-33 of the Abstract) need to be combed through and carefully edited for clarity.
  • Initially, PRKG-AS1 expression did not differ significantly in tissues from patients with gastric cancer (shown in Fig. S9). However, PRKG1-AS1 expression was significantly lower in the group with high 145 expression than in that with low UC.145 expression (based on the median value of UC.145 expression). We thought that this could provide a new way to approach the diversity of gastric cancer, in terms of re-examining the expression of genes that do not have a clear association with gastric cancer. We have corrected the contents of lines 477-481 as follows: “Subsequently, the effect of UC.145 and PRKG1-AS1 interaction on DKK1 expression and Wnt signaling was investigated in AGS and MKN74 cells using western blotting. Suppression of UC.145 increased DKK1 expression, but decreased Wnt5a, MMP7, and β-catenin expression. Notably, regulation of PRKG1-AS1 expression restored changes in gene expression caused by UC.145 (Fig. 5j).”

From these results, we focused on the interaction between UC.145 and PRKG1-AS1, expecting that the two lncRNAs would be involved in different signaling pathways and act as competing endogenous RNAs. Because the experiment was designed and tested for this purpose, the single combination was not included in the figure. We respect the opinions of the reviewers. However, we hope you will understand what led us to present the experimental design and results in Fig. 5j in our study.

Specific comments: 

  1. Line 45: According to the WHO, GC is the fourth leading cause of cancer-related deaths (not third).
  • Thank you very much for this correction. When writing the manuscript, we referenced studies based on data from 2018 and cited the relevant references. We apologize for not updating the manuscript with the latest data. We have revised and updated the text and bibliography.
  1. Source of cell lines: None of the cell lines listed in the Methods sections are on ATCC's website, as indicated on Line 77. Furthermore, all cell lines used in the study and their appropriate culture conditions should be included here.
  • We initially conducted our experiments without recognizing the cross contamination problem between the MKN74 and MKN28 cell lines. To correct this mistake, we replaced MKN28 with the AGS cell line and performed all the experiments again. This study was started in 2012, and the mistake of conducting the study without recognizing the problem with MKN28 is huge. We will do our best not to repeat this mistake again. We also corrected information in "methods" about where exactly the AGS and MKN74 cell lines were purchased. And we put the following link with information about this cell line.

AGS cell line: https://www.atcc.org/products/crl-1739 (ATCC)

MKN74 cell line: https://cellbank.snu.ac.kr/eng/tmpl/sub_main.php?m_cd=46&m_id=0201&sp=2&c_id=559 (Korean cell line bank)

  1. All siRNA sequences used (ie. DKK1, EZH2) should be included in the Methods.

Dkk1, EZH2, PRKG1-AS1

  • Thank you for this suggestion, and we apologize for our omission. All siRNA sequences have been included in the revised manuscript.
  1. Antibody information including manufacturer/source should be added to the Methods.
  • We apologize for not including the antibody information. Supplier information for all antibodies and reagents have been included in the revised manuscript.
  1. Fig 1b and 1d are too small and cannot be read at their current sizes
  • In order to improve the readability of Fig. 1b and 1d, we removed information considered to be somewhat unnecessary. These figure panels are more easily understood in the revised manuscript.
  1. Since the GC + adjacent normal tissue are matched pairs, Fig 1f graph would be better presented as paired data points (in the format of “before-after”) with a paired t-test used for statistical analysis
  • Thank you for suggesting another way to represent the results in Fig. 1f. We have changed this graph according to the reviewer’s suggestion.

Figure 1. Characteristics and expression of UC.145 in gastric cancer (GC). (f) UC.145 expression measured by qRT-PCR for 100 paired GC and normal tissue samples.

  1. What is the meaning of the extra groups of columns in Fig 1g-h graphs with no labels or mention in the text?
  • We apologize for this mistake and the inconvenience caused to the reviewers. Unfortunately, a problem occurred during submission of the manuscript and some figure labels were lost due to compatibility issues with Illustrator. All labels have been restored and the figures have been rearranged so that there are no problems with data interpretation, as shown below.

Figure 1. Characteristics and expression of UC.145 in gastric cancer (GC). (f) UC.145 expression measured by qRT-PCR for 100 paired GC and normal tissue samples; (g) RIP analysis performed with the anti-EZH2 antibody using AGS and MKN74 cell lysates. UC.145 expression in cell lysates and immunoprecipitates measured by qRT-PCR. LncRNA HOTAIR served as the positive control; (h) UC.145 expression in nuclear and cytoplasmic fractions of AGS and MKN74 cells measured by qRT-PCR; (i) Knockdown effect of siRNA on UC.145 expression in AGS and MKN74 cells measured by qRT-PCR; (j) Increased expression of UC.145 upon transfection with overexpression vector in AGS and MKN74 cells measured by qRT-PCR. Data are shown as mean ± SD of three independent experiments (* P ≤ 0.05; ** P ≤ 0.01).

  1. Line 264 is confusing and should be edited for clarity
  • We appreciate the reviewer’s attention to detail. We have modified this sentence to clarify our meaning. We apologize for our error in communicating the results. I think the revised sentence will help you understand our intended meaning: “EZH2 is reportedly involved in the epigenetic regulation of GC [20, 22]. A total of 54 candidates were predicted to bind to EZH2 among the noncoding RNAs obtained from the microarray results, scored using the prediction programs RNAInter, PRIdictor, and RPIseq.”
  1. Fig 3j results cannot be properly interpreted without a siDKK1 alone control. If KD of DKK1 increases proliferation on its own (very possible since it is a Wnt inhibitor), then the combination of siDKK1 and siUC145 is additive rather than “partially restored”
  • Thank you very much for raising a concern we had not previously considered. Based on the reviewer’s suggestion, we confirmed the effect on the proliferation of gastric cancer cells after knocking down DKK1 using siDKK1 alone. As expected by the reviewer, inhibition of DKK1 expression in AGS cells induced a slight increase in cell proliferation. However, this effect was not observed with MKN74 cells. The reason for the discrepancy may be due to differences in DKK1 expression between the two types of cells. Our results indicated that there was almost no DKK1 expression in MKN74 cells compared to that in AGS cells.

Figure 3. UC.145 modulates the methylation of DKK1 in gastric cancer (GC). (j) Effect of DKK1 expression on cell proliferation in AGS and MKN74 cells in which the expression of UC.145 was suppressed [siControl (siCT), siUC.145 (si1, si2) or siDKK1].

  1. Include Westerns to verify KD of EZH2 by siRNAs
  • Thank you for providing this comment. These results are presented in Fig. 4d. Although the expression of EZH2 differed between the two gastric cancer cell lines, we demonstrated that our two siRNAs worked effectively to knock down EZH2 expression.

Figure 4. UC.145 epigenetically regulates DKK1 involved in the canonical Wnt signaling pathway. EZH2-targeting siRNA suppressed DKKI expression at the (c) mRNA and (d) protein levels.

  1. Fig 4i is missing DMSO/GSK126 labels
  •  

  • We apologize that the labels were omitted in Fig. 4i (Fig. 4k in revised manuscript). Unfortunately, several mistakes occurred during the submission process due to compatibility issues with Illustrator. Thank you very much for your detailed review and allowing us the opportunity to correct our mistakes. The labels have been restored in the revised manuscript, as shown below.

Figure 4. UC.145 epigenetically regulates DKK1 involved in the canonical Wnt signaling pathway. (k) Changes in DKK1, EZH2, and H3K27me3 expression regulated by UC.145 were confirmed by western blotting following treatment with EZH2 inhibitor GSK126.

Reviewer 3 Report

Study of Yoon and col. evidenced lncRNA UC.145, lncRNA PRKG1-AS1, and DKK1 relation as a novel prognostic signaling involved in gastric cancer and proposed among these molecule the UC.145 as a better prognostic and diagnostic biomarker   

Major comments : The study is well performed however the authors could make attention to not too emphasized their results with lncRNA UC.145 as a targeted therapy and a practicable important biomarker. Their results although of interest are preliminary and must be underlined as potentially novel biomarkers. Results necessitate much more studies to be validated before as a target for treatment or a biomarker for both diagnosis and prognosis.

Thus :

1) authors must specify them criteria to define UC145 and DKK1 cut-off used for survival analysis in Fig 5.

2) Suvival may be take in consideration at least the TNM stage of cases reported

3) eliminate the concept of targeted and RNA-target therapy (lane 460-461). The main Interest of the study must be focused on the demonstration of the molecular modulation of DKK1 and WNT pathway

4) what is the demonstration to support GC-rich UC145 interacts with the N-terminal helix of EZH2?

5) “pratical” is not an adeguate term, all biomarkers in the biological fluid are easy to obtain. Eliminate or change  this term in lanes 481 and 492

6) authors must underline that results are preliminary and necessitate further studies to confirm their results for application as biomarkers (diagnostic or prognostic?) or as a target for new drug discovery

Author Response

Response to Reviewer 3

Comments and Suggestions for Authors

Study of Yoon and col. evidenced lncRNA UC.145, lncRNA PRKG1-AS1, and DKK1 relation as a novel prognostic signaling involved in gastric cancer and proposed among these molecule the UC.145 as a better prognostic and diagnostic biomarker   

Major comments : The study is well performed however the authors could make attention to not too emphasized their results with lncRNA UC.145 as a targeted therapy and a practicable important biomarker. Their results although of interest are preliminary and must be underlined as potentially novel biomarkers. Results necessitate much more studies to be validated before as a target for treatment or a biomarker for both diagnosis and prognosis.

Thus :

  • authors must specify them criteria to define UC145 and DKK1 cut-off used for survival analysis in Fig 5.
  • Thank you for this suggestion. The cut-off values for survival analysis were based on the average level of UC.145 and DKK1 expression in gastric cancer tissues obtained using the of 2-ΔΔCT The cut-off and hazard ratio values have been included in the survival analysis figure, as shown below.

Figure 5. lncRNA PRKG1-AS1 regulated by UC.145 affects DKK1 expression. (k) Kaplan–Meier estimates of the overall survival rate of patients with high or low UC.145 expression; log-rank P = 0.0116 (left panel). The overall survival rate of patients with high expression of UC.145 and low expression of DKK1 and PRKG1-AS1. Log-rank P = 0.0117 (right panel). Data are shown as mean ± SD of three independent experiments (* P ≤ 0.05; ** P ≤ 0.01).

  • Suvival may be take in consideration at least the TNM stage of cases reported
  • Thank you for proivding this suggestion. We classified the cancer stages using the clinical information provided with the tissue samples from patients with gastric cancer, and we re-analyzed overall survival according to cancer stage. The results indicated that stage III C was most strongly associated with UC.145, DKK1, and PRKG1-AS1. These results are shown in Fig. S10.

Figure S10. Survival stage analysis. Kaplan–Meier estimates of patient overall survival in each cancer stage using the number of cases in which each gene was either upregulated or downregulated.

  • eliminate the concept of targeted and RNA-target therapy (lane 460-461). The main Interest of the study must be focused on the demonstration of the molecular modulation of DKK1 and WNT pathway
  • Thank you for this helpful comment. We have modified the manuscript to focus on the molecular biology approach, rather than the potential therapeutic applications of our research, as recommended by the reviewer.

4) what is the demonstration to support GC-rich UC145 interacts with the N-terminal helix of EZH2?

  • We suggest that the G-rich motif of UC.145 may be important for binding to EZH2. However, further studies are needed to confirm this hypothesis and determine which part of EZH2 interacts with UC.145. In a previously published study, a series of deletion mutations in human PRC2 revealed that the primary N-terminal helix of EZH2 is crucial for RNA binding via a G-rich motif. Given these results, we predict that the G-rich motif contained in UC.145 could also interact with the native N-terminal helix of EZH2 [1,2]. These direct interactions will be investigated in the future.
  • “practical” is not an adequate term, all biomarkers in the biological fluid are easy to obtain. Eliminate or change this term in lines 481 and 492
  • Thank you for your careful review and helpful comments. We have deleted the word 'practical' and modified the sentences accordingly. We apologize for the inappropriate word usage.

6) authors must underline that results are preliminary and necessitate further studies to confirm their results for application as biomarkers (diagnostic or prognostic?) or as a target for new drug discovery

We appreciate the reviewer' comment. We have emphasized in the abstract and conclusion that our study suggests a potential biomarker for gastric cancer and the results warrant further research. These corrections have helped us better articulate our research.

References

  1. Long, Y.; Bolanos, B.; Gong, L.; Liu, W.; Goodrich, K.J.; Yang, X.; Chen, S.; Gooding, A.R.; Maegley, K.A.; Gajiwala, K.S.; et al. Conserved RNA-binding specificity of polycomb repressive complex 2 is achieved by dispersed amino acid patches in EZH2. Elife 2017, 6, doi:10.7554/eLife.31558.
  2. Wang, X.; Goodrich, K.J.; Gooding, A.R.; Naeem, H.; Archer, S.; Paucek, R.D.; Youmans, D.T.; Cech, T.R.; Davidovich, C. Targeting of Polycomb Repressive Complex 2 to RNA by Short Repeats of Consecutive Guanines. Mol Cell 2017, 65, 1056-1067 e1055, doi:10.1016/j.molcel.2017.02.003.
